

# Mass-conserving coupling of total column $CO_2$ ($XCO_2$) from global to mesoscale models: Case study with CMS-Flux inversion system and WRF-Chem (v3.6.1)

Martha P. Butler [1], Thomas Lauvaux *[1], Sha Feng [1], Junjie Liu [2], Kevin W. Bowman [2], and Kenneth J. Davis [1]

[1]Department of Meteorology and Atmospheric Science, The Pennsylvania State University, University Park, PA, USA
[2]Jet Propulsion Laboratory, California Institute of Technology, Pasadena, CA, USA
*Now at Laboratoire des Sciences du Climat et de l'Environnement, CEA, CNRS,UVSQ/IPSL, Universite Paris-Saclay, Orme des Merisiers, 91191 Gif-sur-Yvette cedex, France

**Correspondence:** Thomas Lauvaux (tul5@psu.edu)

**Abstract.** Quantifying the uncertainty of inversion-derived fluxes and attributing the uncertainty to errors in either flux or transport continue to be challenges in the characterization of surface sources and sinks of carbon dioxide ($CO_2$). It is also not clear if fluxes inferred in a coarse-resolution global system will remain optimal in a higher-resolution modeling environment. Here we present an off-line coupling of the mesoscale Weather Research and Forecasting (WRF) model to optimized biogenic $CO_2$

5    fluxes and mole fractions from the global Carbon Monitoring System inversion system (CMS-Flux). The coupling framework consists of methods to constrain the mass of $CO_2$ introduced into WRF, effectively nesting our North American domain within the global model. We test the coupling by simulating Greenhouse gases Observing SATellite (GOSAT) column-averaged dry-air mole fractions ($XCO_2$) over North American for 2010. We find mean model-model differences in summer of ∼0.12 ppm. While 85% of the $XCO_2$ values are due to long-range transport from outside our North American domain, most of the model-

10    model differences appear to be due to transport differences in the fraction of the troposphere below 850 hPa. The framework methods can be used to couple other global model inversion results to WRF for further study using different boundary layer and transport parameterizations.

# 1 Introduction

15    One of the persistent challenges in the study of the global carbon cycle is the quantification of the uncertainty in inferred biogenic carbon sources and sinks (Enting et al., 2012). Contemporary solution methods include inversions using global models and in situ or satellite observations of carbon dioxide ($CO_2$) to correct initial estimates of these biogenic surface fluxes (e.g., Gurney et al., 2002; Baker et al., 2006; Maksyutov et al., 2013). However, in spite of increasing sophistication in the methods, annual inversion surface flux solutions do not agree well at continental scales (Peylin et al., 2013). Contributions to this



disagreement include poor representation of the heterogeneous land surface in relatively coarse global models (Schuh et al., 2010), as well as aggregation and atmospheric transport errors (Kaminski et al., 2001). Inversions using column-averaged $CO_2$ ($XCO_2$) show promise (Liu et al., 2017) as the $XCO_2$ from satellites are presumed to be less susceptible to planetary boundary layer (PBL) atmospheric dynamics and heterogeneous surface fluxes (Rayner and O'Brien, 2001; Keppel-Aleks et al., 2011).

Assimilating column-averaged $CO_2$ observations is not without problems in terms of seasonal global coverage, interference from clouds, and large-scale transport errors (Connor et al., 2016). The density of observations increases to unprecedented levels requiring averaging or thinning techniques so that global inverse models can ingest the large volume of data collected monthly (Liu et al., 2017). An increase in atmospheric model resolution would potentially provide a better representation of the observed spatial variability in $XCO_2$ over continents and allow for the assimilation of individual satellite retrievals (Lauvaux

and Davis, 2014), and may become even more important for the use of observations from geostationary satellites (Polonsky et al., 2014).

Regional models, capable of transporting $CO_2$ as a trace gas, have been used effectively for regional studies (Sarrat et al., 2007; Diaz Isaac et al., 2014) and to generate footprints or back trajectories from observation locations to correct in-domain flux inventories (e.g., Schuh et al., 2010). Several studies illustrated the value of regional models in exploring the near-field

variability of $CO_2$ fluxes and observed mole fractions, using a biogeochemical flux model coupled to a mesoscale atmospheric model and lateral boundary conditions modified from a global model (Ahmadov et al., 2007, 2009; Sarrat et al., 2007). These uses of regional models, however, often assume that the large-scale boundary inflow is sufficiently known so that fluxes within the regional domain dominate the observed variability (e.g., Göckede et al., 2010a, b).To deal with total observable mole fractions of long-lived trace gases, background mole fractions (advected from outside the regional domain) must be dealt

with carefully (e.g., Huang et al., 2010). Various strategies are in use; some are borrowed from the discipline of atmospheric chemistry research: constant concentrations, profiles from aircraft sampling, curtains derived from data products, climatologies or average conditions from global models (see Tang et al. (2007) for a review). For short campaigns, aircraft profile sampling can establish a curtain wall of boundary conditions in the upwind direction. Profile sampling schemes can be used to correct climatological conditions using monthly mean values. These climatologies or average conditions may be used for short-lived

trace gases (e.g., Pfister et al., 2011), but they do not fairly represent varying atmospheric circulation and transport. For long-lived trace gases such as $CO_2$, the vast majority of the molecules in any given volume is the result of long-range transport originating outside the simulation domain.

Long-running regional $CO_2$ studies require time-varying, full-profile lateral boundary conditions. One source of these boundaries has been inversion-optimized mole fractions from global models, with and without adjustment to account for bi-

ases in the global model. For example, Göckede et al. (2010a, b) used 4-D lateral boundary conditions from the CarbonTracker global model (Peters et al., 2007) and verified the high sensitivity of regional inverse fluxes to the $CO_2$ advected at the lateral boundaries. Bias-correcting offsets or adjustments to global model mole fractions have also been made based on comparisons to remote clean-air observations (Schuh et al., 2013). Lauvaux et al. (2012) used a two-step approach, first adjusting the modeled mole fractions with local aircraft profiles and, second, optimizing them within the inversion system. Gourdji et al. (2012)

compared inversion results using an empirical data product derived from Pacific Ocean marine boundary layer observations



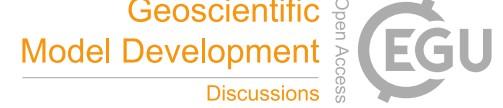

and aircraft profiles (Masarie and Tans, 1995; GLOBALVIEW-CO2, 2013) following Gerbig et al. (2003) and described in Jeong et al. (2013). Ahmadov et al. (2007, 2009) incorporated $CO_2$ initial and lateral boundary conditions from a global model (LMDZ; Peylin et al. (2005)). Their lateral boundary conditions consisted of the results of a forward run of surface fluxes in LMDZ with an added constant offset to adjust the modeled $CO_2$ mole fractions for general agreement with European in situ observations.

In our study, we assume the optimized mole fractions from the global model can be used without adjustment. We also choose the regional domain boundaries to be remote from the main area of interest in our study, and aim to conserve the mass of $CO_2$ introduced at the boundaries to be consistent between the global model and the regional model. This enables us to simulate column-averaged $CO_2$ ($XCO_2$) in both global and regional models and compare to satellite-derived observations. This permits us to explore the impact in the regional model of surface fluxes optimized in the global model. Do these fluxes produce equivalent results in the regional domain? How large are the differences in simulated $XCO_2$ due to the use of different atmospheric transport models? Are the transport errors impairing our ability to infer surface sources and sinks? These are long-standing problems in $CO_2$ studies.

In our experiment, we introduce the optimized biogenic surface fluxes and posterior 4-D atmospheric mole fractions of $CO_2$ from the Carbon Monitoring System (CMS-Flux; Liu et al. (2014)) as initial and boundary conditions into the WRF-Chem regional model (Powers et al., 2017; Skamarock et al., 2005; Grell et al., 2005). Here we describe this framework for achieving our goal of conserving the mass of $CO_2$ introduced in the regional model from the global model. In Section 2, we describe the global and regional models used in this experiment, the model coupling framework, and introduction to the methods for comparison to observation data. We present between-model consistency and comparisons to observations in Section 3. Discussion of the results follows in Section 4. Finally conclusions are presented in Section 5, including how the framework can be used to couple other global models to WRF.

## 2 Models and Methods

In this section, we provide an overview of the CMS-Flux inversion system, the customized WRF-Chem model, the coupling of the two modeling environments, and the observations used to compare model results.

### 2.1 CMS-Flux Inversion System

The NASA CMS-Flux inversion system (http://cmsflux.jpl.nasa.gov) is described in detail in Liu et al. (2014), an observing system simulation experiment (OSSE) to optimize biogenic fluxes using simulated Greenhouse gases Observing SATellite (GOSAT) $XCO_2$ soundings (Yakota et al., 2009). The GOSAT Project (Japan Aerospace Exploration Agency, National Institute of Environmental Studies, and Ministry of the Environment) measures densities of carbon dioxide and methane from space. The $CO_2$ column abundances detected by the Thermal and Near Infrared Sensor for Carbon Observation-Fourier Transform Spectrometer (TANSO-FTS; Yakota et al. (2009)) that we use in our experiment are processed using the NASA Atmospheric $CO_2$ Observations from Space (ACOS) algorithm (Crisp et al., 2012; O'Dell et al., 2012) for use as column-averaged $XCO_2$



observations. The CMS-Flux inversion system uses the forward GEOS-Chem global chemical transport model (Nasser et al., 2010, 2011) driven by meteorological fields from the NASA Goddard Earth Observing System, Version 5 (GEOS-5) data assimilation system (Rienecker et al., 2008). $CO_2$ is simulated as a passive tracer forced by emissions from fossil fuel, biomass burning, shipping and aviation, biogenic land, and ocean surface fluxes. Nasser et al. (2011) used GEOS-Chem as the transport

model in a Bayesian synthesis inversion for surface $CO_2$ fluxes constrained by mid-tropospheric $CO_2$ from the Tropospheric Emission Spectrometer. The GEOS-Chem adjoint model (Henze et al., 2007) optimizes the biogenic surface fluxes and atmospheric $CO_2$ mole fractions to be consistent with $XCO_2$ from a satellite observing system (Liu et al., 2014, 2015). The GEOS-Chem adjoint has been used to estimate carbon monoxide emissions (Kopacz et al., 2009, 2010) and to attribute direct ozone radiative forcing (Bowman and Henze, 2012). Here we use the imposed surface fluxes and the optimized biogenic fluxes

and atmospheric $CO_2$ mole fractions from a coarse resolution global CMS-Flux inversion (horizontal resolution: $4°$ latitude x $5°$ longitude; vertical resolution: 47 levels to 0.01 hPa) that assimilated GOSAT $XCO_2$ for 2010. Details of the imposed and prior fluxes can be found in Liu et al. (2014) and are summarized for North America in Table 1. Figure S1 shows a schematic of the CMS-Flux system and the WRF-CMS interface for our experimental setup.

## 2.2 Weather Research and Forecasting with Chemistry Model (WRF-Chem)

The mesoscale model is WRF-Chem v3.6.1 (Powers et al., 2017; Skamarock et al., 2005; Grell et al., 2005; Fast et al., 2006) with the modification described in Lauvaux et al. (2012) to transport greenhouse gases as passive tracers. Trace gas boundary conditions are provided from a global model at 6-hourly intervals and surface fluxes introduced hourly. This work will describe (Section 2.3 below) the framework for introducing the boundary conditions from the global model into the WRF domain in a manner designed to preserve the vertical distribution of $CO_2$ at the WRF domain edges. Lauvaux et al. (2012) used WRF-Chem

for the forward transport model in an inversion for $CO_2$ fluxes in the USA Upper Midwest region as part of the Mid Continent Intensive (MCI) campaign. This WRF-Chem implementation was also used in Lauvaux and Davis (2014), who investigated the impact of introducing column-averaged $XCO_2$ into regional inversions that typically use only $CO_2$ observations measured in the planetary boundary layer.

In the experiment described here, the regional domain contains most of North America in a Lambert Conformal projection

at 27 km horizontal resolution. The model has 50 levels up to 50 hPa with 20 levels in the lowest 1 km. The model meteorology is initialized every 5 days with 6-hourly ERA-Interim (Dee et al., 2011) reanalysis at T255 horizontal resolution (∼80 km resolution over North America). Each meteorological re-initialization is started at a 12-hour setback from the end of the previous 5-day run. The first twelve hours of the new run are then discarded. We also employ the alternative lake surface temperature initialization as described in the WRF User Guide (Weather Research & Forecasting ARW Version 3 Modeling

System User's Guide, January 2015, pp3–26). Choices of the model physics parameterizations used in this experiment are documented in Table 2.

Carbon dioxide from surface fluxes from the CMS-Flux inversion (optimized biogenic fluxes and imposed surface fluxes from the ocean and emissions of fossil fuel, biomass burning, and ship bunker fuel; Table 1) are carried as individual tracers in WRF-Chem. Background $CO_2$ mole fractions, supplied as boundary conditions from the CMS-Flux optimized mole fractions,



are in a separate tracer. For analyses requiring total $CO_2$ or $XCO_2$, the surface flux tracers are summed, dried of water vapor content, and added to the boundary condition tracer. This multiple-tracer strategy allows for inspection of the separate tracers throughout the model integration. If total column values are required, the region above the 50 hPa top of the WRF model is populated with the appropriate value from 50 hPa to 0.01 hPa from the global model. In this experiment, we do not include

the non-surface fluxes of $CO_2$ from the CMS-Flux inversion (aircraft source and chemical source; Table 1), but these are included in the boundary condition mole fractions from the global domain. A test, reducing these non-surface sources to surface emissions in the WRF domain, made, at most, a 0.03 ppm difference in the simulations of GOSAT $XCO_2$.

The final preparation step for our experiment is the population of the WRF atmosphere with $CO_2$ from the global model. It takes approximately a month's integration in model time to distribute the contributions from the boundaries throughout the

WRF atmosphere. For this experiment, we started the model integration in early December 2009, but begin all analyses in January 2010.

### 2.3   Mass Conserving Coupling Framework

In this experiment, we wish to compare simulated column-averaged $CO_2$ from both the WRF execution and the CMS-Flux products with the ACOS GOSATv3.5 soundings (O'Dell et al., 2012) for North America. This requires that we ensure that the

mass of $CO_2$ introduced into the WRF domain from the CMS-Flux GEOS-Chem global model be conserved, both at the surface (fluxes) and at the boundaries (atmospheric mole fractions) of the WRF model domain. The challenges for mass conservation include differences in model horizontal grid resolution, implied grid surface elevation, and vertical grid discretization. The strategy described here can also be used to couple the WRF regional model to other global models with a minimal amount of customization specific to the global model.

### 2.3.1   Mass Conservation of Surface Fluxes

We apply domain-wide scaling factors for each surface flux to conserve the mass of $CO_2$ introduced into the domain by the surface fluxes. First, we create a map projection for translation from the CMS-Flux GEOS-Chem $4° \times 5°$ grid to the WRF 27 km grid (Figure S2 illustrates the domain extent of the WRF Lambert Conformal projection). After assignment of WRF grid cells to the global grid, we compute monthly scaling factors for each surface flux as follows:

1. Calculate the sum of the mass exchange in the global model grid cells assigned to the WRF domain.

2. Calculate an initial domain-wide sum of mass exchange for the WRF grid cells using the assigned global model grid cells.

3. Compute a domain-wide scaling factor as the ratio of the results of step 1 and step 2.

4. Multiply the mass exchange assigned to each WRF grid cell by the domain-wide scaling factor.

These steps are repeated for each of the surface fluxes. The goal is to achieve equal total surface mass exchanges at the hourly resolution of the flux input into the WRF model domain. The component fluxes from CMS-Flux have temporal resolutions





varying from hourly to monthly. The optimized biogenic flux is at monthly resolution, but with a 3-hour diurnal cycle overlay with monthly net zero emission/uptake. We do not scale the diurnal cycle overlay. Most monthly scaling factors are in the range [1.0, 1.3] with the exception of some of the minor fluxes (biofuel and ship bunker fuel) with slightly larger scaling factors. The scaled surface fluxes are introduced into the WRF modeling system using the WRFCHEMI function of WRF-Chem. Realistic

flux scaling results may also be achieved using a more sophisticated approach, such as the mass-conserving utilities within NCL ESMF (http://www.ncl.ucar.edu/Applications/ESMF.shtml), which would also retain the global model flux patterns, as well as preserving the domain mean.

### 2.3.2   Mass Conservation at the Domain Boundaries

The challenge in the case of the domain boundaries is not only the difference in horizontal resolution and grid type, but also the

difference in vertical discretization schemes. WRF uses a terrain-following, hydrostatic-pressure vertical eta coordinate with a fixed model top. The $CO_2$ boundary mole fractions for our experiment are from a GEOS-Chem model with 47 hybrid-sigma layers corresponding to the GEOS-5 (MERRA) reduced vertical grid (Rienecker et al., 2008). This grid is terrain-following from the surface up to ~170 hPa, with fixed pressure levels from 170 hPa to the top of the model. Depending on the surface pressure and terrain, there may be 9 layers below 1 km, compared to 20 layers below 1 km for our WRF domain. To approximate

the vertical distribution of $CO_2$ in the global model source in the WRF grid, we follow these steps for each WRF boundary grid cell:

1.  Compute an equivalent global model surface pressure for each WRF boundary grid cell using standard bi-linear interpolation (http://numerical.recipes; Interpolating in two or more dimensions) using the four global model surface grid cells whose centers are closest in latitude and longitude to the WRF grid cell center.

2.  Compute the equivalent global model pressure column using this derived surface pressure and the standard vertical grid discretization for the global model, in this case the GEOS-5 hybrid sigma-pressure $ap$ and $bp$ parameters and algorithm for the 47-layer reduced vertical grid (http://wiki.seas.harvard.edu/geos-chem/index.php/GEOS-Chem_vertical_grids).

3.  Compute an equivalent WRF pressure column using this derived surface pressure and the $znu$ WRF vertical resolution discretization vector.

4.  Assign a source global model level for each receiving WRF model level in the WRF boundary grid (Figure 1). This is determined by the global model level edge pressures between which the derived WRF midpoint layer pressure falls. This computation is done in log space with the respective pressure columns in units of Pa.

5.  Use simple linear interpolation (if necessary) between global model levels to smooth out a poorly mixed flux signal. We used this technique for CMS-Flux GEOS-Chem where the first four or five model levels in WRF are sourced from the

first GEOS-Chem level; this source layer often shows the immediate result of the surface flux as distinct from several well-mixed layers above the surface layer.




6. The result of this transfer of vertical $CO_2$ columns from the global model to the WRF boundary grid cells is introduced into WRF via the WRFCHEMBC functionality used for meteorological boundary conditions, similar to the approach of (Ahmadov et al., 2007, 2009).

We use the GEOS-Chem surface pressure for two reasons. First, it is the mass in the GEOS-Chem column that we want to introduce into the WRF model domain, and, second, it is not possible to match the surface pressures between the two models due to different horizontal grid resolutions, model grid surface elevations, and driver meteorology. To test the adequacy of this method, we compute pressure-weighted column-averaged $XCO_2$ along the WRF boundaries, independently compute the same quantity from the global model up to 50 hPa, and compare the results. An example of this comparison for a day in early June 2010 is shown in Figure 2. The surface layer of the western and eastern boundaries of the WRF domain is predominantly ocean, where we do not expect significant model grid elevation or surface pressure differences. This is evident for this example date in Figure 2b. On the other hand, we do expect some differences in the southern and northern boundaries, particularly in mountainous areas (Figure 2a). Our algorithm produces model-model differences of column-averaged $CO_2$ at the boundaries of less than 0.1 ppm on most of the boundaries and less than 0.3 ppm in the high-terrain regions of the northern boundary.

### 2.4 Model Comparison to Observations

While a primary goal of this model coupling experiment is to compare WRF and CMS-Flux model-simulated $XCO_2$ GOSAT soundings, it is comparisons to other observations that provide verification and insight into model behavior. In addition to the GOSAT $XCO_2$, we compare to $XCO_2$ from a Total Carbon Column Observing Network (TCCON) site for times of GOSAT overpasses. We also compare model meteorological winds to observations from selected North American rawinsonde sites.

### 2.4.1 GOSAT $XCO_2$

We simulate in both the CMS-Flux and WRF model atmospheres more than 13,000 good quality ACOSv3.5 LITE GOSAT soundings (http://co2.jpl.nasa.gov) within the WRF North American domain during 2010. The distribution of these soundings varies in space and time, with no coverage in ocean areas or north of $60°N$ in winter. We follow the same sampling scheme in both WRF and CMS-Flux model $CO_2$ mole fractions:

1. Locate the nearest model grid cell in time and space to the GOSAT sounding.

2. Isolate the column of $CO_2$ at that grid cell, calculate the dry air mole fractions.

3. Interpolate the simulated $CO_2$ to the 20 pressure levels specified in the sounding's ACOSv3.5 averaging kernel algorithm and a priori profile (O'Dell et al., 2012; Rodgers and Connor, 2003) as shown in the ACOS 3.5 User Guide.

4. Apply the averaging kernel to the interpolated profile for the full column $XCO_2$ or use the a priori profile pressure-level weights for partial column analysis.

Using the pressure-level weights versus the complete averaging kernel for the whole column yields results within a few tenths of a ppm for the full columns in our model atmospheres. For a priori profile pressure levels above 50 hPa, we use the CMS-Flux





interpolated optimized mole fractions for both the WRF and CMS $XCO_2$ computations. For ACOS GOSAT soundings with profile pressure levels below the model surface, we use the $CO_2$ at the midpoint of the surface model layer.

### 2.4.2   Lamont TCCON $XCO_2$

The Total Carbon Column Observing Network (TCCON; Wunch et al. (2010)) provides measurements of $XCO_2$ from the
earth's surface at a global network of sites. Two TCCON sites within our North American domain were operating in 2010. The Park Falls site has significant drop-outs, especially in summer, so we choose the Lamont TCCON site (36.6°N, 97.49°W) for comparison (Wennberg et al., 2017). We average GOSAT soundings in a box of 6° latitude and 12° longitude centered at the Lamont site and match them to the TCCON data averaged for the hour of the GOSAT overpass of this regional box. We also computed hourly-averaged simulated $XCO_2$ from the models for these GOSAT soundings to compare to both GOSAT
and TCCON $XCO_2$. During our period of comparison, summer of 2010, this region covers a large gradient of surface $CO_2$ fluxes. Our goal is not to evaluate the GOSAT $XCO_2$ relative to TCCON observations, but rather to illustrate how well the model-simulated $XCO_2$ follow the general tendencies of the observations. More rigorous evaluation of the GOSAT $XCO_2$ would require use of coincidence criteria as shown in Wunch et al. (2011) or Basu et al. (2013).

### 2.4.3   Horizontal Wind at Selected Rawinsonde Sites

We also make use of the NOAA archive of rawinsonde data (http://www.esrl.noaa.gov/raobs/fsl-format-new.cgi; Schwartz and Govett (1992)) for model comparisons to observed wind speed and direction at mandatory reporting levels. The goal in this case is to identify possible causes of transport differences between models and observations. Comparisons of 00 UTC soundings (16-19 LST, depending on the site) are summarized to show annual and seasonal biases.

## 3   Results

### 3.1   Evaluation of Transport Differences in Model-Simulated GOSAT $XCO_2$ Soundings

We compare the CMS-Flux and WRF $XCO_2$ model simulations with each other in Figure 3 and with the GOSAT soundings in Table 3. If we have reached our target of conserving $CO_2$ mass entering the WRF domain from the CMS-Flux optimized fluxes and boundary conditions, then we expect the model-simulated $XCO_2$ to be similar. Model-model differences can be related to transport (horizontal transit times from different driver meteorology and vertical mixing from boundary layer processes) or
model resolution (heterogeneous surface characteristics). The seasonal mean model-model differences in $XCO_2$ simulations are largest in winter (mean, -0.30 ppm; RMSD, 0.51 ppm) and smallest in summer (mean, 0.12 ppm; RMSD, 0.72 ppm). These seasonal distributions of differences are all sharply peaked around zero with a few large outliers (>5 ppm) in summer. Comparisons of model simulations to the GOSAT soundings are summarized in Table 3. In spite of a few extreme outliers in winter and spring, the inner quartile ranges (IQRs) for both models relative to GOSAT are approximately 2 ppm, with nearly
the same seasonal mean differences (CMS-Flux range [0.00, 0.31] ppm; WRF range [-0.04, 0.38] ppm). Median differences are



slightly larger than the mean differences, except for the CMS-Flux simulations in summer and the WRF simulations in summer and fall. Although the model results are centered well with the GOSAT soundings, neither model produces simulations with the full range of values of the GOSAT $XCO_2$. We might have expected more variability in the WRF simulations, as multiple GOSAT soundings assigned to a single grid cell in GEOS-Chem are represented by many grid cells in the higher resolution

WRF grid. However, we do not see any differences at this summary level. There are six individual GOSAT $XCO_2$ soundings with model-GOSAT differences greater than 10 ppm; these are all in challenging terrain in the western United States.

These general comparisons, however, do not show differences in spatial coverage by season of the GOSAT $XCO_2$ soundings or any spatio-temporal variations in model-model residuals. To address this, we aggregate the model-model differences to the CMS-Flux GEOS-Chem $4°$ x $5°$ grid to map the mean seasonal differences in each grid cell (Figure 4). We report in Figure

4 only those grid cells with more than 10 GOSAT soundings during each season shown. There are consistent small negative residuals in the WRF $XCO_2$ simulations relative to CMS-Flux in most of the continent in winter and in the south and east in fall. The WRF – CMS-Flux differences are slightly positive in the northwest in fall. The pattern in spring is mixed. There are strong positive differences in the Pacific Northwest in summer. The underlying CMS-Flux optimized biogenic flux shows a very strong source of $CO_2$ in the upper Pacific Northwest, and a correspondingly strong sink in the MidWest and East in July.

These sources and sinks are evident in the WRF model layers closest to the surface, but not always in the CMS-Flux optimized mole fractions in the surface layers in the same locations. On some days, the surface layer flux contributions in the CMS-Flux mole fractions are in grid cells adjacent to the emitting grid cells, suggesting some model-model differences in mixing and transport within the boundary layer. There are also sharp gradients in terrain height in this region that may contribute to the model differences, although this does not appear to be an issue in the Rocky Mountain region of the US West. In general, there

is more variability in the spatial differences during the growing season. These maps also show how the spatial distribution of the GOSAT soundings changes by season. Note that there are no ocean soundings and no soundings in the high latitudes in winter. Examination of variances of the model-simulated soundings on this spatial scale show no clear differences between models, other than that there is less spatial variance in the models than in the GOSAT soundings.

Having documented the model-model differences in column-averaged $XCO_2$ at seasonal scales, we next compare the distri-

bution within the columns of the CMS-Flux and WRF $XCO_2$ simulations. We expect to find the most differences in the active growing season, so focus the comparison on summer. We divide the columns into upper and lower portions, with the lower column corresponding to the three pressure levels closest to the surface in the ACOS algorithm, and with the upper column consisting of the remaining 17 levels. The ACOS averaging kernel pressure level weights specify ∼12.5% of the column-averaged value from the three pressure levels closest to the surface. The elevation above the surface of this split varies with

terrain, but roughly corresponds to 850 hPa. As justification for this division, we highlight an example $CO_2$ profile at the location of the LEF tall tower in Park Falls, Wisconsin, USA ($45.95°$N, $90.27°$W) at the time of a GOSAT overpass on 27 August, 2010 (Figure 5). In this example, the $XCO_2$ simulations from both the WRF total $CO_2$ and the CMS-Flux optimized mole fractions agree with each other and with the ACOS GOSAT $XCO_2$ (Figure 5a). The GOSAT $XCO_2$ value is 385.526 ppm with an uncertainty of 1.075 ppm; WRF and CMS-Flux simulated values are 385.706 and 385.245 ppm, respectively. We

decompose the WRF total column $CO_2$ into contributions from the global model (light blue boundary conditions profile in



Figure 5a) and the contributions of the flux tracers (Figure 5b). This is a location far from the boundaries of the WRF regional domain, so the boundary conditions profile is the result of long-range transport, not recent inflow. At this location and time, the flux contributions to the total $CO_2$ profile are predominantly below 850 hPa. The minor fluxes (ocean, and combined biomass burning, biofuel, and ship bunker fuel emissions) contribute very little to the overall column value. The biogenic flux and the

imposed fossil fuel emissions constitute almost all the flux portion of the column value. The transported boundary conditions account for more than 85% of the column-averaged $CO_2$. For applications using total column $CO_2$, it is important that this coupling of regional to global model be done correctly.

We illustrate the model-model differences for the simulated GOSAT $XCO_2$ in summer using this ~850 hPa division in Figure 6. Differences shown are for sub-column-averaged $CO_2$ within the lower and upper portions of the column, computed

as the sum of the interpolated mole fractions at each ACOS pressure level times the ACOS-provided weight at that pressure level. The lower portion of the column accounts for ~50 ppm of the total column-averaged $CO_2$. Comparing spatial patterns of differences of the sub-columns versus the total column in Figure 4c, we see that the WRF positive residual in the Pacific Northwest is the result of larger mole fractions in both upper and lower sub-columns. This is an area with a relatively low count of GOSAT soundings (Figure S3) and very large biogenic source flux. In the eastern half of North America, there is a dipole

effect, with WRF simulations having lower mole fractions in the lower sub-column and higher values in the upper sub-column compared to the CMS-Flux simulations. There are non-homogeneous patches of strong uptake in the biogenic flux in eastern North America in July, generally matching the spatial pattern in the lower sub-column. The upper sub-column pattern more closely resembles the total column pattern in Figure 4c. Model-model differences in other seasons are minimal in both parts of the column. The summer patterns illustrate that agreement of full column $XCO_2$ simulations does not necessarily imply that

the distribution of the $CO_2$ is the same in the two models. This further suggests model-model differences in transport within the columns, possibly due to more vertical mixing in the CMS-Flux GEOS-Chem model. The CMS-Flux vertical profile in Figure 5a shows an example of this behavior.

## 3.2    Temporal Evaluation of Model-Simulated $XCO_2$ at the Lamont TCCON Site

An independent comparison of column-averaged $XCO_2$ for GOSAT and the model simulations is possible using the $XCO_2$

observations at the Lamont TCCON site. GOSAT passes near the Lamont TCCON site around 19 UTC. We select the GOSAT soundings near Lamont, average the $XCO_2$ values by day and report them along with the 19 UTC hourly average of the TCCON observations. We present the weighted averages and standard deviations of both sets of observations in Figure 7a. The error bars in Figure 7a represent the root mean square uncertainties of the selected TCCON and GOSAT soundings included in the hourly averages. In the example shown for days in July and August 2010 when both observations are available, there

are 1–21 good GOSAT soundings and 3–34 TCCON $XCO_2$ observations during the 19 UTC overpass. Despite the presence of outliers in the early summer (around DOY 180–190), daily average GOSAT and TCCON $XCO_2$ soundings agree within 1–2 ppm, with discrepancies likely due to sampling and representation errors. The mean residuals or ACOS GOSAT, CMS-Flux, and WRF $XCO_2$ relative to TCCON for this period are $0.088 \pm 1.856$ ppm, $0.868 \pm 1.042$ ppm, and $0.965 \pm 0.981$ ppm, respectively (Figure 7b). The residuals from the model simulations are generally, but not exclusively, positive and are more





similar to each other than to either of the observations. This implies that model transport differences are small compared to other model-data differences at this time and location. The synoptic-scale variability in the summer atmospheric $CO_2$ mole fractions is well represented during short-term events (e.g., near DOY 218). Despite the coarser resolution of the CMS-Flux mole fractions compared to WRF, both models are able to capture the trend and variability observed by the Lamont TCCON

site.

### 3.3 Horizontal Mean Wind

We selected a group of rawinsonde sites (Table 4; Figure 8) including west coast, east coast, Gulf Coast, and mid-continent North American sites for comparison of model and observed wind speed and wind direction at mandatory rawinsonde reporting levels. Our intent is to identify any regional distinctions in biases or variability that might help to explain the $CO_2$ model-data

differences. Each of these sites had more than 11 months of data for 00 UTC soundings in 2010 at the mandatory reporting levels of 925 hPa, 850 hPa, 500 hPa and 250 hPa which we used in our analysis. Continental mountain and high plains sites were not included because they lack 925 hPa data, and cannot easily be compared with mandatory levels at lower elevation sites. Model locations were assigned as the nearest grid cell with similar surface height. For some coastal sites, the representative model sites were moved one grid cell toward the ocean to achieve this. In Figure 9, we show the annual mean wind speed

bias at these sites from the WRF forward run (initialized with the ERA-Interim reanalysis) and the GEOS-5 reanalysis used by the CMS-Flux system. Recall that although the WRF run is initialized with reanalysis data, it is free-running during each 5-day run, and so may not conform to the reanalysis. Wind speed bias, RMSE, variance ratio and correlation skill score ($\frac{Cov(model,obs)}{\sqrt{Var(model)Var(obs)}}$; von Storch and Zwiers (1999)) are summarized in Table 5 for mandatory reporting level 925 hPa. (See Tables S1-S3 for results at other reporting levels.) In general, coastal sites are difficult to simulate in both models, with seasonal

differences that tend to cancel each other in WRF but not in the GEOS-5 meteorology. With the exception of the coastal sites, WRF wind speed error is positively biased at the lower levels, but this bias largely disappears with height. Most notable is that GEOS-5 underestimates the wind speed for nearly all of this selection of sites at all levels. The WRF model overestimates wind speed variability by more than 10% at 925 hPa and 850 hPa, but slightly underestimates variability at higher mandatory levels. The GEOS-5 analysis underestimates variability consistently at all levels. Both models' correlation skill scores improve with

height, as expected (Table 5 and Tables S1-S3). WRF overestimates day-to-day variability, as shown by the variance ratio, in winds at lower levels, and has a larger RMSE than GEOS-5 at lower, but not higher, levels. GEOS-5 underestimates day-to-day variability at all levels. Annual wind direction bias and RMSE, for both models at all sites and mandatory levels are smaller than the number of degrees separating reportable wind directions (not shown). However, at the 925 hPa mandatory level there are seasonal directional biases at many sites in summer and at west coast sites in all seasons (Figure 10). This directional

bias may contribute to the summer boundary layer differences in $XCO_2$ simulations between the two models. Based on these results, the potential improvement from finer resolution WRF simulations in the resolution of mesoscale features and vertical structure near the surface is not evident from our continental-scale evaluations. We discuss further in Section 4 the impact of potential biases in WRF near the surface and in GEOS-5 at upper levels.





## 4 Discussion

Based on the spatiotemporal distribution of GOSAT $XCO_2$ soundings, there is very good agreement between the two modeling systems and GOSAT in the gross seasonal comparisons, well within the uncertainty of the individual GOSAT retrievals (0.5–2.00 ppm, reported with each sounding). Our primary goal of this framework is to control the $CO_2$ mole fractions introduced

into the WRF domain to be as close as possible to the mole fractions from the global model. By scaling the surface fluxes and by constraining the mole fractions in the flow at the boundary walls, we have approached this goal of mass conservation. The largest spatiotemporal differences in $XCO_2$ shown in this experiment appear to be due to the model-model differences in transport of the anomalous July biogenic source in the continental northwest and offsetting sinks in the midwest and east in the optimized CMS-Flux biogenic fluxes. However, the very small differences in simulated $XCO_2$ values mask larger differences

within the columns. For example, we can see model-model differences in the transport of the surface flux anomaly in the Pacific Northwest in summer, both within and above the boundary layer (Figure 6). Apparent model differences in vertical mixing within the boundary layer in summer in the eastern part of North America result in lower values within the boundary layer (-0.5 to +0.1 ppm) and higher values above the boundary layer (by +0.1 to +0.5 ppm) in the mesoscale model compared to the global model. We know that this configuration of WRF-Chem lacks convective mass transport of the $CO_2$ tracers. This

will affect the vertical transport of $CO_2$ into and out of the boundary layer. The CMS-Flux optimzied $CO_2$ mole fractions also show effects of recent fluxes in the surface layer, and then nearly homogeneous mixing in the next several model layers, as seen in the example profile in Figure 5a.

In this experiment, the meteorological evaluation of the WRF results was approximately comparable to the GEOS-Chem GEOS-5 results, and neither compared well in all respects with the rawinsonde observations. At the selected sites, the GEOS-

5 winds were slow by up to 3 m s$^{-1}$ at nearly all the sites and mandatory levels we used for comparison, while the WRF winds were closer to observations at higher levels. We had hoped to see better performance from the increased resolution, both horizontal and vertical, in WRF. Perhaps the horizontal resolution in WRF is still too coarse to take advantage of any truly mesoscale effects. We also did not assimilate meteorological observations into WRF; this was by design as the primary focus of the experiment was to identify differences in transport of the $CO_2$ originating from the CMS-Flux optimized biogenic fluxes.

The WRF resolution, the assimilation of meteorological data in WRF, and changes to the boundary layer parameterizations within WRF could be tested to review the conclusions we see here.

One of the main objectives of satellite $XCO_2$ programs is to provide observations for assimilation into atmospheric inversions to improve the quality of inferred surface fluxes. The satellites provide good coverage of the North American domain in the summer, the season with the most biogenic activity. However, coverage in other seasons is limited, and must be supple-

mented with other $CO_2$ observations, forcing the challenge of assimilating both column and surface observations in the same inversion. It is an interesting thought experiment to see what the model-model differences would be if we did have complete satellite sampling coverage over the course of a year. We sampled each model at its own horizontal grid resolution at the hour of the day with the most GOSAT $XCO_2$ soundings (20 UTC), created simulated $XCO_2$ from $CO_2$ columns interpolated to the pressure levels commonly used in the ACOS algorithm, and examined model-model differences in the same way as we did with





the simulated GOSAT soundings. We compared the differences for summer, the season with the best GOSAT spatial coverage (See Figures S4–S6). WRF values below 850 hPa are lower than the CMS-Flux values in the east, and WRF values are higher over most of the continent above 850 hPa. These results are reasonably consistent with the summer results shown in Figures 4 and 6, which are conditioned on the spatiotemporal coverage of the GOSAT soundings. This is encouraging. While we will

never have perfect coverage in every season with a satellite $XCO_2$ product, the results presented here show that model-model differences do not appear to be overly dependent on the GOSAT sampling coverage.

We have created a viable framework for comparing the transport of surface fluxes optimized in one model in another model. Theoretically, this allows for comparisons using different grid resolutions, meteorological drivers, and model parameterization options. In this experiment, we do see some differences between models in horizontal winds and boundary layer mixing, but

see no clear advantage of the mesoscale model in the simulation of the satellite-derived $XCO_2$. As a caveat, this experiment used one combination of parameterizations of boundary layer processes in the mesoscale model. The framework does provide a computationally feasible laboratory for investigating other possible model configurations. Repeating the experiment in WRF in other configurations (boundary layer physics parameterizations, driver meteorological fields, assimilation of observed meteorology, or finer model resolutions) could yield an ensemble of results that might better establish an envelope of transport

uncertainty for the CMS-Flux optimized biogenic flux solution.

## 5  Conclusions

We have established a framework for effectively nesting a mesoscale model within a global model achieving approximate mass conservation of the trace gas $CO_2$. The CMS-Flux and WRF simulated column-averaged GOSAT $XCO_2$ samples are comparable within a few parts per million with some spatial differences, across all seasons and all geographic locations in the

WRF North American domain. The models used here have very different horizontal and vertical resolutions and computational grids. Scaling the surface fluxes appropriately could be done with a readily-available mass-conserving regridding utility rather than using the custom-computed scaling factors used in this experiment. The code for the provision of the boundary conditions from the global model to the mesoscale model is available from https://github.com/psu-inversion. There is a version for TM5-based models such as CarbonTracker (https://www.esrl.noaa.gov/gmd/ccgg/carbontracker), and versions can be created with

straightforward modification for other global models with rectangular grids. This framework makes it possible to follow the $CO_2$ from surface fluxes optimized in a global model into a regional domain, allowing for testing of various transport options. Hourly WRF 3D $CO_2$ fields for 2010 from this experiment are archived at the ORNL DAAC (Lauvaux and Butler, 2016) for community use.

Although simulations of entire columns differ very little between the models in this experiment, there are vertical differences

in the distribution of the $CO_2$ within the column, likely attributable to differences in mixing within the two model regimes. These differences, insignificant when considering total column $XCO_2$ from the models, will affect the inverse sources and sinks, attributing signals to the surface and the boundaries differently for each model. Unfortunately, both models show significant biases with respect to meteorological observations that vary with season. At the sites we examined, wind speeds are


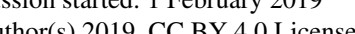


low in GEOS-5 in all seasons, especially at higher altitudes. WRF shows positive biases near the surface. There is the potential for significant transport bias with either modeling system, and in this experiment, WRF is not obviously better. Assimilating meteorological observations in WRF should reduce the wind bias in WRF, but the vertical mixing requires modification of the physical parameterizations. As also found by Diaz Isaac et al. (2014) when comparing to CarbonTracker, WRF appears to

amplify $CO_2$ fluctuations in the boundary layer more than is found in the CMS-Flux optimized mole fractions.

*Code and data availability.* The ACOS-GOSAT v3.5 soundings were produced by the ACOS/OCO2 project at the Jet Propulsion Laboratory, California Institute of Technology from GOSAT column $CO_2$ spectra, and were obtained from http://co2.jpl.nasa.gov. The ACOS v3.5 User Guide is also available at this site. CMS-Flux system products are available from J. Liu (Junjie.Liu@jpl.nasa.gov). Other data sources are cited in the text and references. The hourly WRF $CO_2$ atmospheric mole fractions are archived as Lauvaux, T. and Butler, M.: CMS:

Hourly Carbon Dioxide Estimated Using the WRF Model, North America, 2010 (http://dx.doi.org/10.3334/ORNLDAAC/1338). Code for the boundary conditions interface to WRF-Chem is available at https://github.com/psu-inversion/WRF_Boundary_Coupling.

*Author contributions.* All authors designed the experiment. M.P.B. conducted the experiment and the initial analysis and wrote the first draft of the paper. M.P.B. and T.L. created the boundary conditions software. M.P.B. and S.F. tested the boundary conditions software. J.L. provided the CMS-Flux system products. All authors contributed to the analysis and the writing of the paper.

*Competing interests.* The authors declare that they have no competing interests.

*Acknowledgements.* This work was funded by the NASA Carbon Monitoring System (https://carbon.nasa.gov) project: Quantification of the sensitivity of NASA CMS-Flux inversions to uncertainty in atmospheric transport (grant NNX13AP34G). Partial support for M.P.B., T.L. and S.F. was also provided by ACT-America (Earth Venture Suborbital grant NNZ15AG76G). Resources supporting the CMS-Flux system products used in this work were provided by the NASA High-End Computing (HEC) Program through the NASA Advanced Supercomputing

(NAS) Division at Ames Research Center. We thank P.O. Wennberg, D. Wunch, C. Roehl and G. Toon for making the Lamont TCCON data available.



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





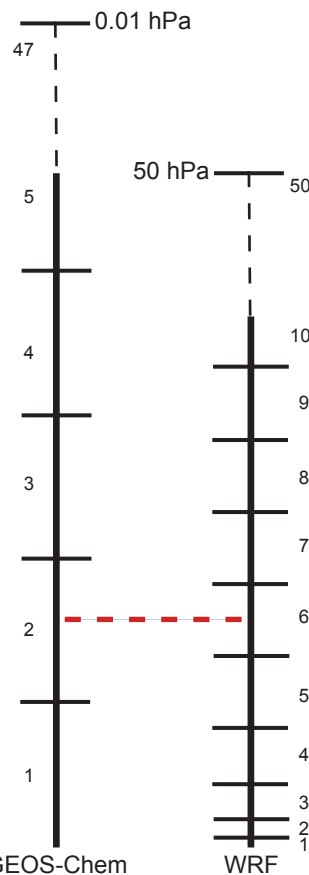

**Figure 1.** Conceptual illustration of the vertical mapping scheme between an example grid cell from the CMS-Flux GEOS-Chem 47-level grid (left) and the corresponding domain boundary grid cell in the WRF-Chem 50-level grid (right). Numbers indicate the levels in each model. In this case, the mass at the pressure midpoint in level 6 of the WRF column is matched to level 2 in the GEOS-Chem column. With no additional interpolation, level 6 $CO_2$ in the WRF column will be sourced from level 2 in GEOS-Chem (see text).





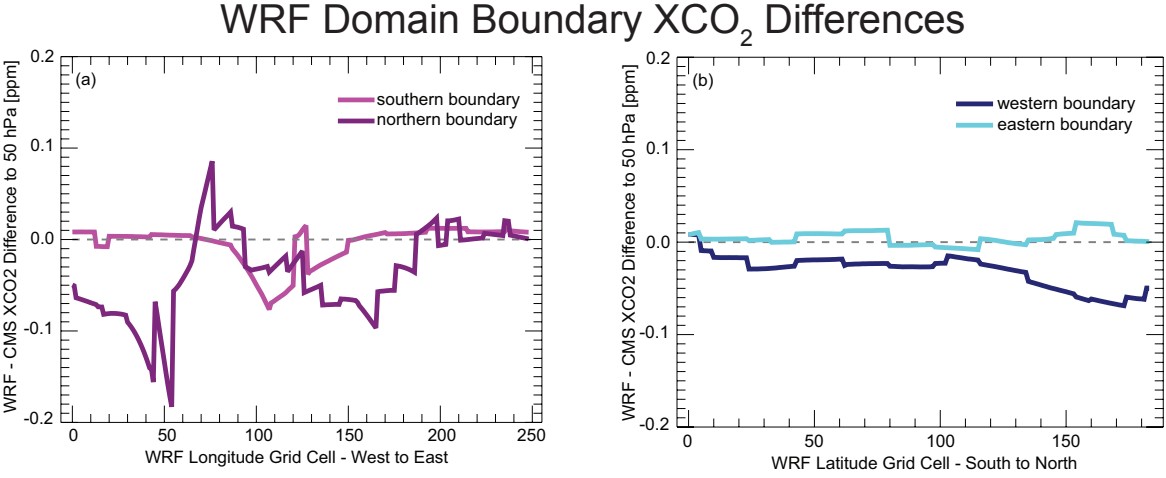

**Figure 2.** Differences in column-averaged $CO_2$ (to 50 hPa) of WRF boundary grid cells compared to the source CMS-Flux GEOS-Chem grid cells (up to 48 hPa). The southern and northern boundaries (a) are mostly over land, illustrating the effects of grid surface and surface pressure differences. The western and eastern boundaries (b) are over ocean except for the northern end of the western boundary.





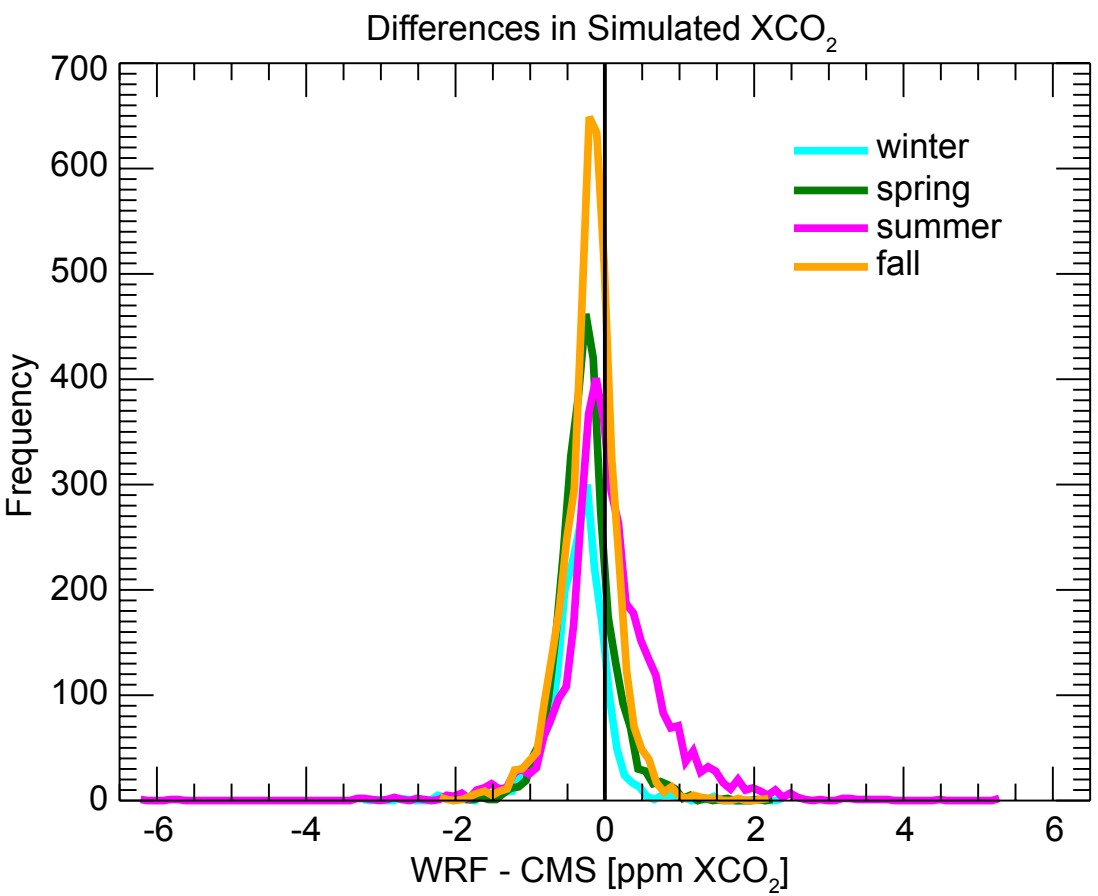

**Figure 3.** Seasonal model-model differences (WRF - CMS-Flux) in simulations of GOSAT $XCO_2$.





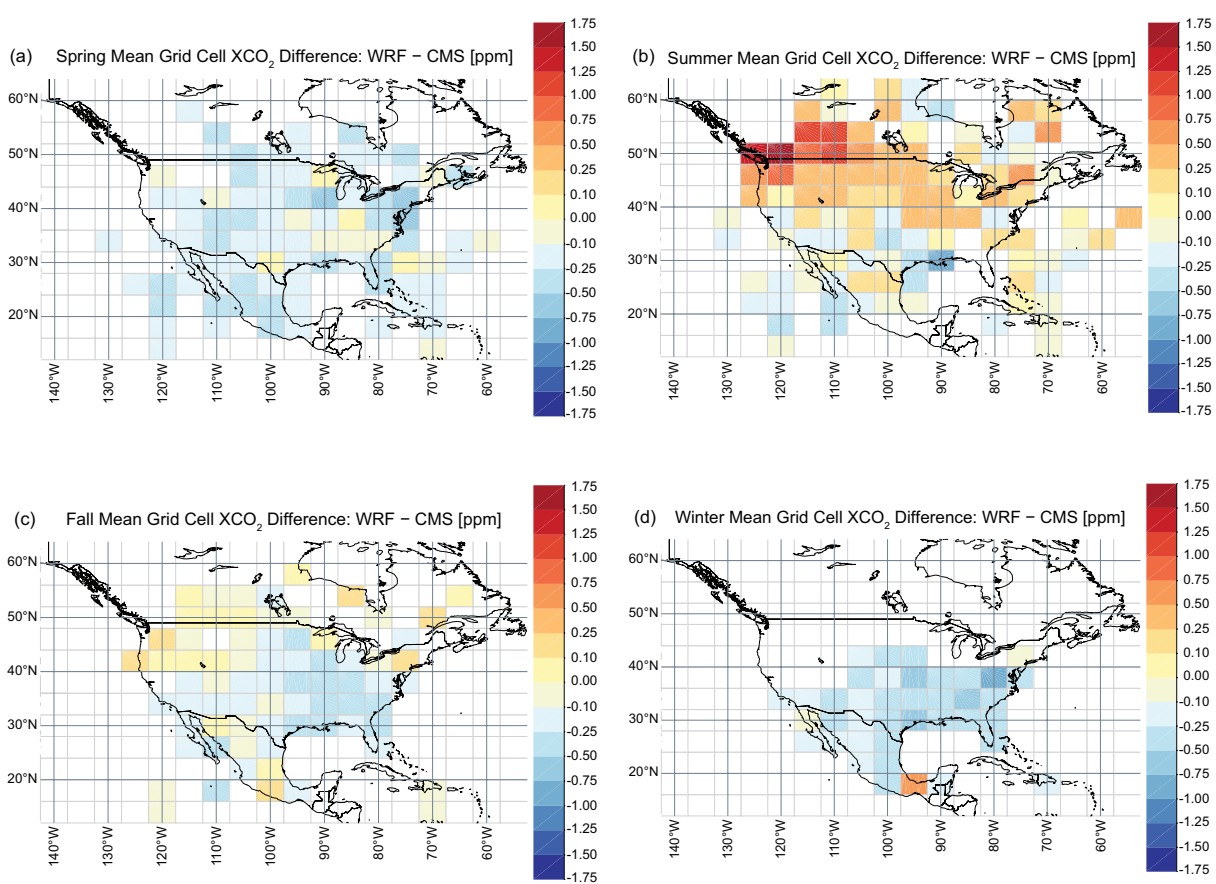

**Figure 4.** Seasonal mean spatial differences in WRF and CMS simulated $XCO_2$ aggregated to the GEOS-Chem CMS-Flux grid (light gray lines) for (a) spring, (b) summer, (c) fall and (d) winter. Grid cells with no shading have fewer than 10 GOSAT soundings for the season indicated.





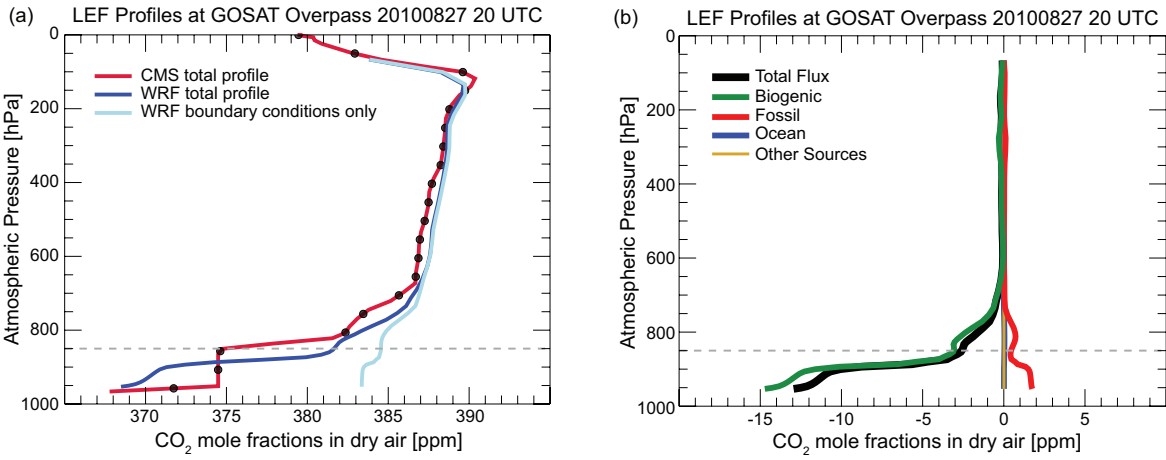

**Figure 5.** Profiles of modeled distribution of $CO_2$ within a column at the LEF tall tower in northern Wisconsin, USA, at the time of a GOSAT overpass on 27 August, 2010. (a) Total $CO_2$ from the CMS-optimized mole fractions and the WRF simulation. Black dots on the CMS profile indicate the 20 pressure levels from the ACOS GOSAT a priori profile. The light blue profile is the transported boundary condition tracer from the WRF simulation. (b) Profiles of the flux tracers from the WRF simulation. The total of the flux tracers and the transported boundary conditions equals the total WRF profile. Other sources include biomass burning, biofuel burning and ship bunker fuel emissions. The dashed line marks the separation of the profiles into lower and upper portions.





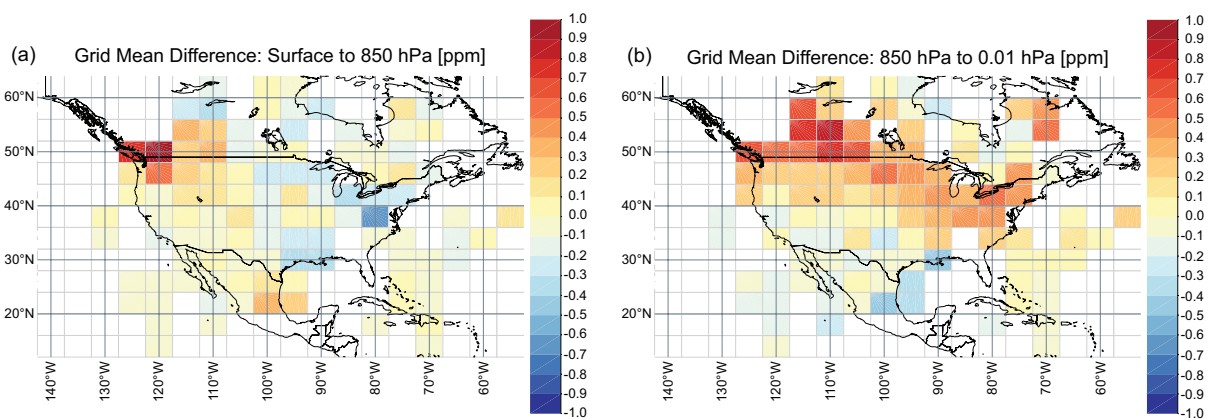

**Figure 6.** Mean summer (June, July, August) spatial differences in WRF and CMS-Flux pressure-weighted $CO_2$ for partial columns of simulated GOSAT $XCO_2$ aggregated to the CMS-Flux grid (light gray lines). (a) Lower portion of column (model surface to 850 hPa) (b) Upper portion of column (850 hPa to top of the ACOS profile).



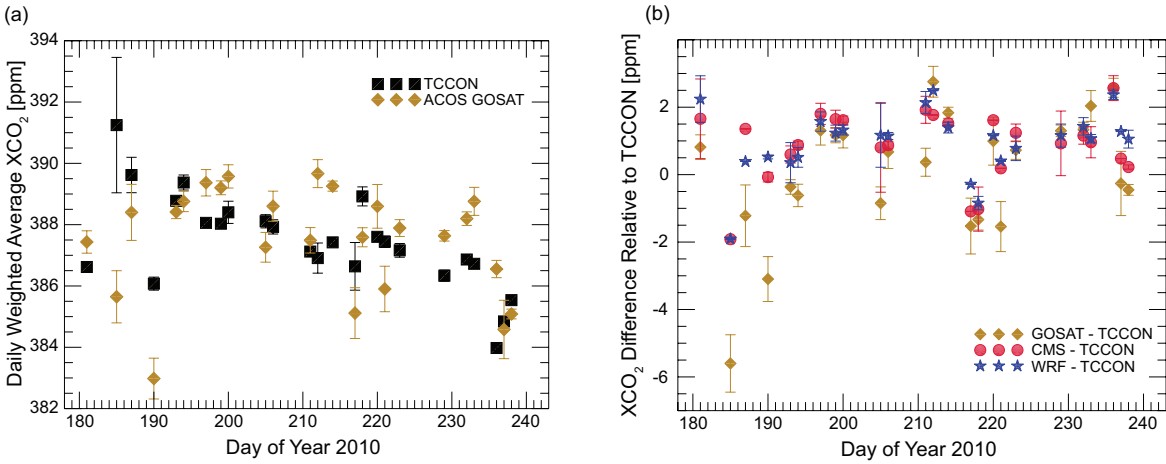

**Figure 7.** (a) Comparison of $XCO_2$ from GOSAT and the Lamont TCCON site in summer 2010. TCCON observations are weighted means and standard deviations during the hour of the GOSAT overpass. GOSAT observations are weighted means of soundings near the Lamont TCCON site. (b) Residuals with respect to the TCCON values for the ACOS GOSAT soundings and the modeled $XCO_2$ simulations from CMS-Flux and WRF.





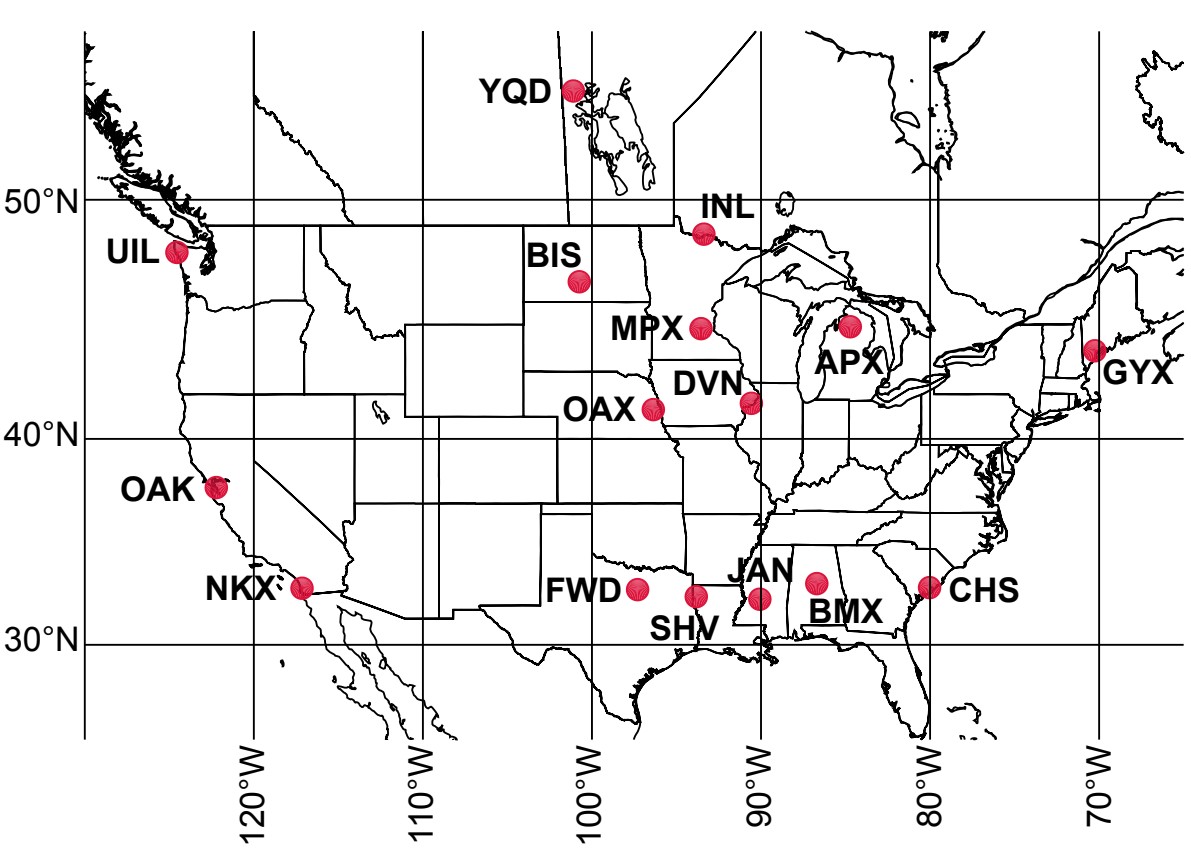

**Figure 8.** Selected rawinsonde sites used for model comparisons to 00 UTC horizontal wind observations.





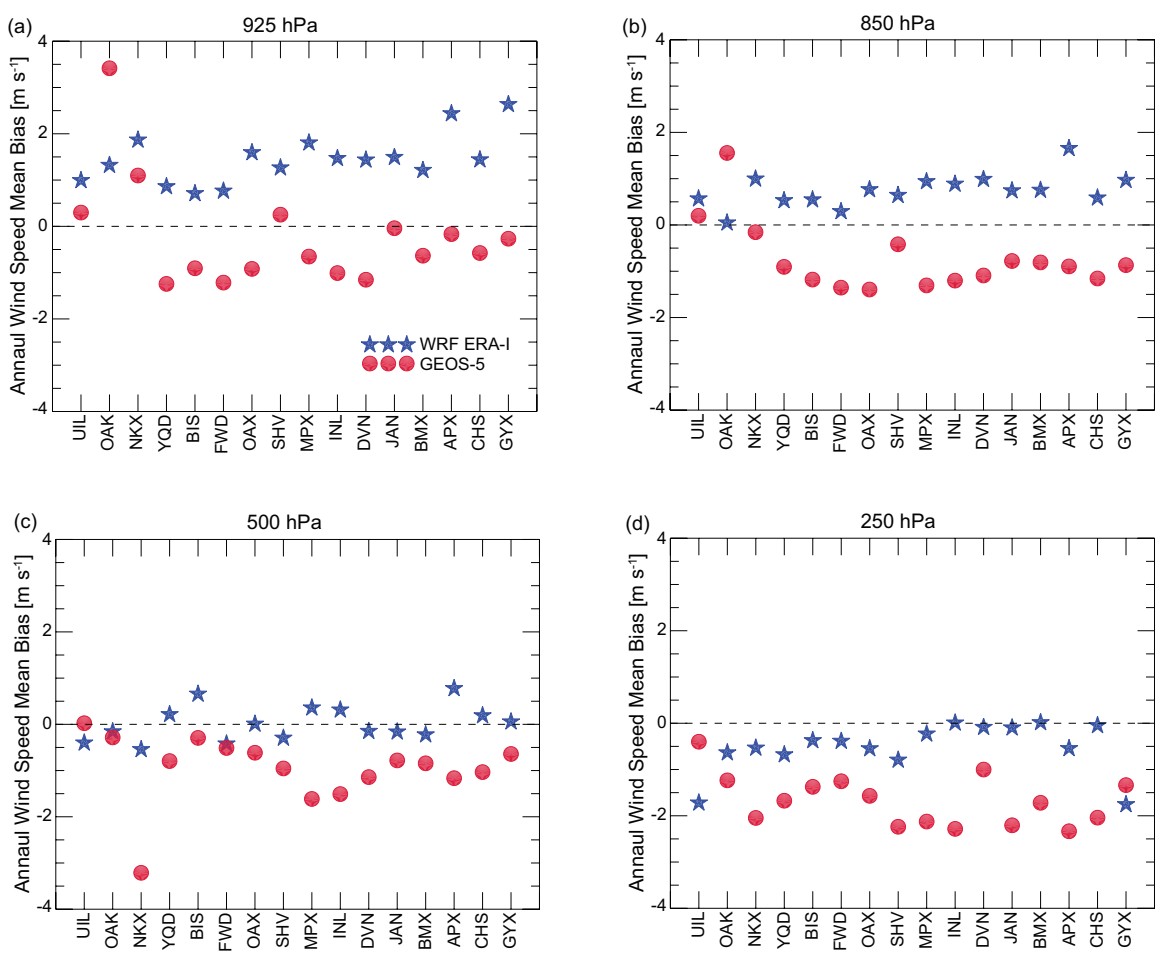

**Figure 9.** Annual mean wind speed bias [m s$^{-1}$] at selected rawinsonde sites for 00 UTC soundings in 2010 for WRF initialized with ERA-Interim reanalysis and for GEOS-5 reanalysis at selected mandatory reporting levels: (a) 925 hPa, (b) 850 hPa, (c) 500 hPa and (d) 250 hPa



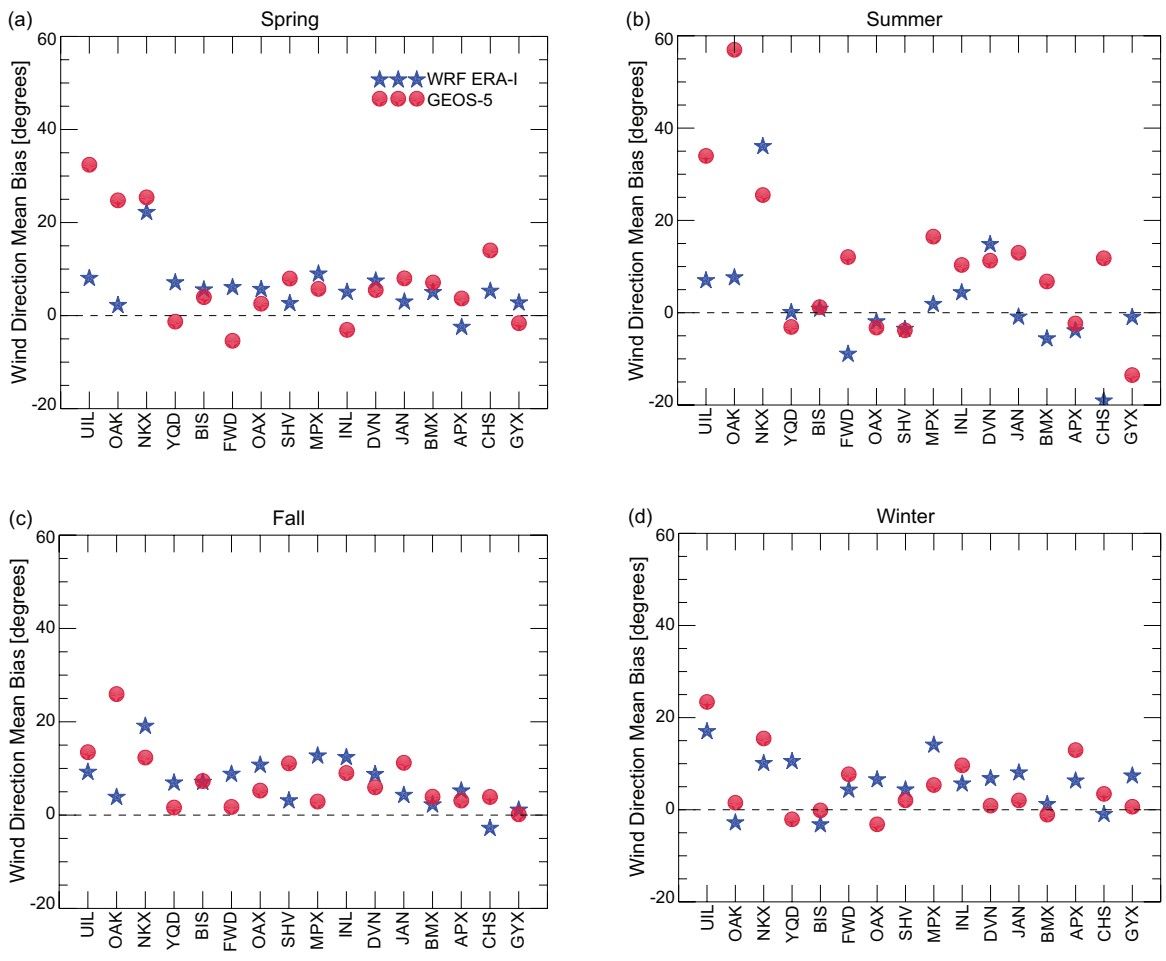

**Figure 10.** Seasonal mean wind direction bias [degrees] at selected rawinsonde sites for 00 UTC soundings in 2010 for WRF initialized with ERA-Interim reanalysis and for GEOS-5 reanalysis at 925 hPa for: (a) spring, (b) summer, (c) fall and (d) winter.



**Table 1.** Surface fluxes for the North American domain in this study

| Surface Flux | Source | Temporal Resolution | Annual Budget (GtC yr$^{-1}$) | Reference |
|---|---|---|---|---|
| Fossil Fuel | CDIAC | Hourly | 1.77 | Andres et al. (2011) |
| Ocean | MITgcm, ECCO2, Darwin project | 3-Hourly | -0.25 | Marshall et al. (1997a, b), Menemenlis et al. (2005, 2008), Follows et al. (2007), Dutkiewicz et al. (2009), Follows and Dutkiewicz (2011) |
| Biogenic | CMS-Flux, optimized from CASA-GFED3 | Monthly, with imposed diurnal cycle | -1.07 | Liu et al. (2014), van der Werf et al. (2004, 2006, 2010), Olson and Randerson (2004) |
| Biomass Burning | GFED3 | Daily | 0.11 | van der Werf et al. (2010), Mu et al. (2011) |
| Biofuel | GFED3 | Monthly | 0.02 | van der Werf et al. (2010), Mu et al. (2011) |
| Ship Bunker Fuel | ICOADS | Monthly | 0.04 | Nasser et al. (2010, 2011) |
| *Aircraft and *Chemical Source | | Monthly | 0.20 | Nasser et al. (2010) |

*These non-surface sources are not included in the results presented in this work.

When released as surface sources, they add $\leq 0.03$ ppm to the XCO$_2$ reported here.

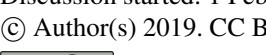



**Table 2.** WRF model physics parameterization choices

| Parameterization | Option Used | References |
| --- | --- | --- |
| Longwave | Rapid Radiative Transfer Model | Mlawer et al. (1997) |
| Shortwave | Dudhia | Dudhia (1989) |
| PBL | Mellor-Yamada-Nakanishi-Niino Level 2.5 | Nakanishi and Niino (2004, 2006) |
| Surface layer | Mellor-Yamada-Nakanishi-Niino | Nakanishi and Niino (2004, 2006) |
| Land Surface Model | Noah | Chen and Dudhia (2001) |
| Cumulus | Kain-Fritsch | Kain (2004) |
| Microphysics | WRF Single Moment 5-class | Hong et al. (2004) |





**Table 3.** Model residuals [ppm] relative to ACOSv3.5 GOSAT $XCO_2$. The span between the distribution quantiles Q3 and Q1 defines the interquartile range (IQR). IQRs for residuals of both models and all seasons are in the range [1.8, 1.9] ppm.

|         | Winter | | Spring | | Summer | | Fall | |
|---------|--------|--------|--------|--------|--------|--------|--------|--------|
|         | CMS | WRF | CMS | WRF | CMS | WRF | CMS | WRF |
| Maximum | 5.91 | 5.51 | 6.52 | 7.45 | 6.58 | 7.31 | 9.42 | 9.03 |
| Q3      | 1.25 | 0.97 | 1.28 | 1.09 | 1.18 | 1.31 | 0.96 | 0.79 |
| Median  | 0.29 | -0.00 | 0.38 | 0.16 | 0.26 | 0.34 | 0.07 | -0.10 |
| Mean    | 0.25 | -0.04 | 0.31 | 0.11 | 0.26 | 0.38 | -0.00 | -0.15 |
| Q1      | -0.63 | -0.91 | -0.58 | -0.79 | -0.61 | -0.52 | -0.87 | -1.01 |
| Minimum | -18.00 | -18.13 | -19.21 | -19.73 | -7.95 | -7.23 | -9.64 | -9.70 |
| Soundings | 1998 | | 3113 | | 3965 | | 4305 | |



**Table 4.** Rawinsonde sites used in comparison of 00 UTC modeled horizontal winds

| Site | WMO ID | Latitude (°N) | Longitude (°W) | Location Name |
|------|--------|---------------|----------------|---------------|
| UIL | 72797 | 47.95 | 124.55 | Quillayut, WA |
| OAK | 72493 | 37.75 | 122.22 | Oakland, CA |
| NKX | 72293 | 32.87 | 117.15 | Miramar, CA |
| YQD | 71867 | 53.97 | 101.10 | The Pas, MB |
| BIS | 72764 | 46,77 | 100.75 | Bismarck, ND |
| FWD | 72249 | 32.80 | 97.30 | Fort Worth, TX |
| OAX | 72558 | 41.32 | 96.37 | Valley, NE |
| SHV | 72248 | 32.45 | 93.83 | Shreveport, LA |
| MPX | 72649 | 44.83 | 93.55 | Chanhassen, MN |
| INL | 72747 | 48.67 | 93.38 | International Falls, MN |
| DVN | 74455 | 41.60 | 90.57 | Davenport, IA |
| JAN | 72235 | 32.32 | 90.07 | Jackson, MS |
| BMX | 72230 | 33.10 | 86.70 | Shelby County, AL |
| APX | 72634 | 44.91 | 84.72 | Gaylord, MI |
| CHS | 72208 | 32.90 | 80.03 | Charleston, SC |
| GYX | 74389 | 41.60 | 70.25 | Gray, ME |



**Table 5.** Model comparison to rawinsonde wind speed at mandatory reporting level 925 hPa, 00 UTC in 2010

| Site | Count | WRF ERA-Interim | | | | GEOS-Chem GEOS-5 | | | |
|---|---|---|---|---|---|---|---|---|---|
| | | Bias | RMSE | Correlation | Variance | Bias | RMSE | Correlation | Variance |
| | | $[ms^{-1}]$ | $[ms^{-1}]$ | Skill | Ratio | $[ms^{-1}]$ | $[ms^{-1}]$ | Skill | Ratio |
| UIL | 350 | 0.99 | 3.41 | 0.87 | 1.09 | 0.30 | 3.41 | 0.84 | 0.67 |
| OAK | 350 | 1.32 | 2.98 | 0.78 | 1.67 | 3.41 | 5.41 | 0.46 | 1.84 |
| NKX | 348 | 1.87 | 4.08 | 0.44 | 1.84 | 1.10 | 2.99 | 0.47 | 0.82 |
| YQD | 363 | 0.86 | 3.36 | 0.76 | 1.12 | -1.24 | 2.67 | 0.86 | 0.68 |
| BIS | 349 | 0.71 | 3.04 | 0.73 | 1.02 | -0.91 | 2.58 | 0.80 | 0.79 |
| FWD | 363 | 0.76 | 3.49 | 0.71 | 1.27 | -1.21 | 3.06 | 0.75 | 0.79 |
| OAX | 364 | 1.59 | 4.30 | 0.71 | 1.13 | -0.92 | 3.08 | 0.81 | 0.72 |
| SHV | 359 | 1.27 | 3.77 | 0.69 | 1.43 | 0.25 | 3.00 | 0.71 | 0.85 |
| MPX | 359 | 1.81 | 3.99 | 0.75 | 1.38 | -0.65 | 3.35 | 0.71 | 0.79 |
| INL | 360 | 1.47 | 3.39 | 0.78 | 1.26 | -1.01 | 3.14 | 0.73 | 0.70 |
| DVN | 357 | 1.44 | 4.10 | 0.76 | 1.25 | -1.15 | 2.85 | 0.86 | 0.69 |
| JAN | 362 | 1.49 | 3.49 | 0.77 | 1.51 | -0.04 | 2.45 | 0.80 | 0.85 |
| BMX | 362 | 1.21 | 3.32 | 0.78 | 1.37 | -0.63 | 2.67 | 0.79 | 0.64 |
| APX | 358 | 2.44 | 4.15 | 0.75 | 1.87 | -0.17 | 2.33 | 0.81 | 1.07 |
| CHS | 352 | 1.44 | 4.20 | 0.67 | 1.30 | -0.58 | 2.47 | 0.84 | 0.72 |
| GYX | 362 | 2.63 | 5.54 | 0.71 | 1.54 | -0.27 | 3.65 | 0.77 | 0.84 |