# Peer review of "Mass-conserving coupling of total column $CO_2$ (XCO2) from global to mesoscale models: Case study with CMS-Flux inversion system and WRF-Chem (v3.6.1)"

_Geoscientific Model Development, 2018_

## Referee Comment (RC1) · Anonymous Referee #1 · 14 Mar 2019

**General Comments**

The authors have created a method for interpolating global model mixing ratios at coarse spatial scales to the grid for higher resolution models such as WRF-Chem, and then evaluate the differences between the resultant simulations with identical surface fluxes against TCCON and co-located GOSAT soundings near the TCCON site for the concentrations and rawinsonde data for the winds. This activity is useful and interesting for the community of regional tracer modelers, but the conclusions don't seem to demonstrate the utility of the boundary condition interpolation technique, and they

certainly don't imply the added value of the regional modeling approach. While this negative result is in itself important, some further evaluation is required to understand why this is the case before I can recommend publication of these results. Specifically, evaluation against aircraft data to better understand the model-model differences in the tracer distribution in the vertical dimension, and evaluation against GOSAT data spatially. The tremendous amount of NOAA surface and tower data would also be extremely useful for differentiating between the PBL dynamics, as would meteorological analysis of the PBL differences between the two models. This represents an expanded scope for the manuscript, but the introduction of the boundary interpolation alone does not represent a significant scientific advance of sufficient scale to warrant publication.

2.3.1 I wonder how sensitive your results are for different diurnal cycles. Particularly at higher spatiotemporal resolution, this could be important for matching observations. It does simplify the interpretation vs. the parent model, though.

2.3.2 To call the method "mass conserving" would suggest that the XCO2 values at the boundaries should be much closer than 0.1ppm, right? I understand that the propagation into the domain might lead to these differences, but I would expect the overall differences at the time of interpolation to be tiny. Perhaps you could demonstrate that the actual mass is conserved, even if the column average mixing ratio is not (due to different surface pressures). Maybe some of the mass is lost in the upper 50hPa? Later on it says that you are using the CMS-Flux mixing ratios above 50mb, which makes this difference even more confusing.

Section 3 It seems that a lot of insight could be gained from comparisons to GOSAT in a spatial context, rather than just the model-model differences at simulated GOSAT sounding locations and times. Why is this not shown? Certainly the comparison in 3.2 at the Lamont TCCON is part of this, but the spatial information could shed light on the boundary condition effects, etc in other parts of the domain. This is reinforced by the very small bias of GOSAT at the Lamont TCCON site relative to the models. Remember that 0.5ppm difference in the column (particularly for a large scale average

as shown in Figure 4) can amount to a significant difference that would be interpreted as a flux difference in an atmospheric inversion.

Further insight into the vertical mixing differences could (and should) be gained by comparison to the NOAA light aircraft time series, for example at Lamont. Since the fluxes are the same in each model, the only difference would be the transport.

Section 4 This would benefit from a comparison with the results recently made available in Schuh et al (2019), in which the authors examine the differences in GEOS-Chem (which drives CMS-Flux) and TM5 (which uses a different reanalysis). In particular, the authors look at the differences in vertical mixing and try attribute these differences to the way convection is handled in a rough way. They also draw conclusions about broad scale flux inference from these differences. It is a complementary study that deserves some mention here.

---

## Referee Comment (RC2) · Anonymous Referee #2 · 10 Apr 2019

**General remarks**

The authors compare total column CO$_2$ over Norther America computed from WRF-Chem with results from the CMS-Flux inversion system which is based on GEOS-Chem. For this purpose they developed a scheme for nesting WRF-Chem into GEOS-Chem in a way that the mass of CO$_2$ introduced into WRF-Chem from GEOS-Chem is conserved. Although only minor differences between the results of the two models were found for the total vertical column of CO$_2$, more pronounced differences were found between the vertical CO$_2$ distributions computed by the two models.

This investigation could generally be useful for the community who do inversions of satellite derived $CO_2$ concentrations. However, the paper suffers in major parts from imprecise language and unclear descriptions of important aspects. In particular, the description of the methods does not allow finding out whether the very little added value of the higher spatial resolution could eventually be attributed to the way how surface $CO_2$ fluxes are implemented in WRF-Chem. I cannot recommend the publication of this paper unless the paper is improved in these aspects.

**Detailed comments**

Throughout the paper the language of the paper is imprecise which makes major parts of the paper hard to read. Already the abstract is a good example: What is meant by 'fluxes' in the first sentence? Fluxes from the surface? Fluxes from which sources? What is meant by 'transport'? Long rage transport? What is 'our' North American domain?

A few further examples (by far not complete) are: Page 1, last line: 'do not agree well': With what? Page 2, lines 21 and 23: The expression 'curtains' is somewhat odd. Page 6, first line: What does 'dried of water vapor' mean here (odd wording anyway)?

The meteorological driver of GEOS-Chem is GEOS-5 while meteorological boundary conditions for WRF-Chem are from ERA interim. In how far can differences between these meteorological drivers contribute to the differences in upper air wind fields and $CO_2$ concentrations from WRF-Chem and GEOS-Chem. This should be analyzed in more detail.

The authors describe in much detail how they achieve mass conservation when deriving $CO_2$ surface fluxes for the WRF-Chem simulation from the fluxes applied to GEOS-Chem. Does this result in surface fluxes smoothed to the GEOS-Chem grid, which are used as input for WRF-Chem? What $CO_2$ emission patterns (anthropogenic, biogenic) are still resolved in the WRF-Chem simulation? It is not clear whether the emissions for the WRF-Chem simulations are really better spatially resolved than for

the GEOS-Chem simulations. Eventually show emisssion input for WRF-Chem and for GEOS-Chem in the supplementary material.

Page 6, line 2: 'We do not scale the diurnal cycle overlay.'? How large is the difference between the diurnal cycles of the emissions applied to GEOS-Chem and WRF-Chem? Section 2.4.2: Why is the averaging performed over such a big area?

Page 9, lines 3-6: These sentences are hard to understand. This should be explained in more detail. Please mention also the magnitude of the differences between the individual GOSAT $XCO_2$ soundings which are located within a single GEOS-Chem cell?

Page 12, line 14: Why is convective transport of $CO_2$ not included? WRF-Chem can handle convective vertical transport of atmospheric trace compounds.

The conclusions must be extended. Currently they contain mostly a description of data availability and a short summary.

---

## Author Comment (AC2) · 31 Oct 2019

**Response to** Anonymous Referee # 1

**"Mass-conserving coupling of total column CO2Åǎ(XCO2) from global to mesoscale models: Case study with CMS-Flux inversion system and WRF-Chem (v3.6.1)" [gmd-2018-342], Butler et al.**

**General Remarks:**

We greatly appreciate the thoughtful and constructive suggestions from **Anonymous Referee # 1.** We have addressed all of the comments and made the revised manuscript clearer. Point-by-point responses follow. The original comments from the reviewer are in italics and the response in normal.

**General Comments:**

*The authors have created a method for interpolating global model mixing ratios at coarse spatial scales to the grid for higher resolution models such as WRF-Chem, and then evaluate the differences between the resultant simulations with identical surface fluxes against TCCON and co-located GOSAT soundings near the TCCON site for the concentrations and rawinsonde data for the winds. This activity is useful and interesting for the community of regional tracer modelers, but the conclusions don't seem to demonstrate the utility of the boundary condition interpolation technique, and they certainly don't imply the added value of the regional modeling approach. While this negative result is in itself important, some further evaluation is required to understand why this is the case before I can recommend publication of these results. Specifically, evaluation against aircraft data to better understand the model-model differences in the tracer distribution in the vertical dimension, and evaluation against GOSAT data spatially. The tremendous amount of NOAA surface and tower data would also be extremely useful for differentiating between the PBL dynamics, as would meteorological analysis of the PBL differences between the two models. This represents an expanded scope for the manuscript, but the introduction of the boundary interpolation alone does not represent a significant scientific advance of sufficient scale to warrant publication.*

**Response:** We thank the reviewer for the valuable recommendations. We have extended significantly our analysis to include i) a comparison with the original coupling scheme, which constitutes a commonly-used approach to couple global to regional models. This section highlights the importance of the boundary interpolation and mass-conservation techniques used here. We also included ii) a comparison to GOSAT data to evaluate the impact of transport on model-data residuals, iii) a comparison to NOAA aircraft profile data to identify differences in the vertical distribution of $CO_2$ mole fractions, and iv) A comparison to tower measurements over the whole year to identify seasonal differences and the relationship between tower-based model-data residuals and column-integrated residuals. These significant additions have been inserted in the main text (Methods, Results, and Discussion sections) and our conclusions were modified accordingly. To support our text, five figures and one table were added to provide a full analysis of the model $CO_2$ mole fractions using various types of measurements.

*2.3.1 I wonder how sensitive your results are for different diurnal cycles. Particularly at higher spatiotemporal resolution, this could be important for matching observations. It does simplify the interpretation vs. the parent model, though.*

**Response**: We agree with the reviewer that diurnal cycles can impact significantly the simulated column and in situ $CO_2$ mole fractions. In this study, we focused on comparing transport models at two different spatial scales using identical surface fluxes. Because CMS is a well-established system, we avoided comparing different diurnal cycles but future studies should account for mis-representation of the short-term variability in surface fluxes. We have added a sentence in the Discussion section to highlight the importance of diurnal variations in simulating atmospheric $CO_2$ mole fractions. "We also acknowledge here that diurnal variations in CMS surface fluxes are prescribed and might not exactly match the meteorological conditions in WRF. Comparison of fluxes

and PBL variations are shorter timescales is needed in future studies."

*2.3.2 To call the method* "*mass conserving*" *would suggest that the XCO2 values at the boundaries should be much closer than 0.1ppm, right? I understand that the propagation into the domain might lead to these differences, but I would expect the overall differences at the time of interpolation to be tiny. Perhaps you could demonstrate that the actual mass is conserved, even if the column average mixing ratio is not (due to different surface pressures). Maybe some of the mass is lost in the upper 50hPa? Later on it says that you are using the CMS-Flux mixing ratios above 50mb, which makes this difference even more confusing.*

**Response**: We show the differences at the boundaries in Fig. 2. The median for the western, eastern, and southern boundaries is about 0.03ppm. The northern boundary shows larger differences due to mountains (0.05ppm on average). Physically, the only approach to reduce further the mass differences is to avoid mountains at the boundaries of the simulation domain. Because the model surfaces are different, extrapolation (or removal) of CO2 is unavoidable. Because our scheme is intended to be used with in situ data, we minimized the modification of mixing ratios while conserving the column mass as best as possible.

For the air mass over 50hPa (not simulated in WRF), we used the CMS values in all our analysis. We clarified that point in the text. "Because the top of the atmosphere in WRF is at 50hPa, we used CMS mole fractions to complete the column values above 50hPa in our study."

*Section 3 It seems that a lot of insight could be gained from comparisons to GOSAT in a spatial context, rather than just the model-model differences at simulated GOSAT sounding locations and times. Why is this not shown? Certainly the comparison in*

*3.2 at the Lamont TCCON is part of this, but the spatial information could shed light on the boundary condition effects, etc in other parts of the domain. This is reinforced by the very small bias of GOSAT at the Lamont TCCON site relative to the models. Remember that 0.5ppm difference in the column (particularly for a large scale average as shown in Figure 4) can amount to a significant difference that would be interpreted as a flux difference in an atmospheric inversion. Further insight into the vertical mixing differences could (and should) be gained by comparison to the NOAA light aircraft time series, for example at Lamont. Since the fluxes are the same in each model, the only difference would be the transport.*

**Response**: This part has now been included with significant additions compared to the original study. We have added comparisons to GOSAT and aircraft profiles to our study. Tower data have also been used to understand seasonal differences.

*Section 4 This would benefit from a comparison with the results recently made available in Schuh et al (2019), in which the authors examine the differences in GEOS-Chem (which drives CMS-Flux) and TM5 (which uses a different reanalysis). In particular, the authors look at the differences in vertical mixing and try attribute these differences to the way convection is handled in a rough way. They also draw conclusions about broad scale flux inference from these differences. It is a complementary study that deserves some mention here.*

**Response**: Since Schuh et al. (2019) has been published after our study was submitted, we didn't refer to it. We can now cite Schuh et al. (2019) and discuss the potential role of deep convection. We have added in the Discussion section "A recent study showed the role of deep convection at Mid-latitudes by comparing two global models coupled to the same surface fluxes (Schuh et al., 2019). Their results suggest that the transport of continental surface fluxes by latitudinal atmospheric transport can greatly

impact the distribution of CO2 mole fractions across the northern hemisphere. Similar to our results, they conclude that additional evaluation of vertical mixing is needed to reduce transport errors above the PBL, esp. by deep convection and other detrainment processes.

---

## Author Comment (AC3) · 31 Oct 2019

**Response to** Anonymous Referee # 2

**"Mass-conserving coupling of total column CO2 ă(XCO2) from global to mesoscale models: Case study with CMS-Flux inversion system and WRF-Chem (v3.6.1)" [gmd-2018-342], Butler et al.**

**General Remarks:**

We greatly appreciate the thoughtful and construc-
tive suggestions from **Anonymous Referee** $\#$ **2.**
We have addressed all of the comments and made the revised manuscript clearer.
Point-by-point responses follow. The original comments from the reviewer are in italics
and the response in normal.

**General remarks**

*The authors compare total column CO2 over North America computed from WRFChem
with results from the CMS-Flux inversion system which is based on GEOSChem. For
this purpose they developed a scheme for nesting WRF-Chem into GEOSChem in a
way that the mass of CO2 introduced into WRF-Chem from GEOS-Chem is conserved.
Although only minor differences between the results of the two models were found for
the total vertical column of CO2, more pronounced differences were found between the
vertical CO2 distributions computed by the two models.*

*This investigation could generally be useful for the community who do inversions of
satellite derived CO2 concentrations. However, the paper suffers in major parts from
imprecise language and unclear descriptions of important aspects. In particular, the
description of the methods does not allow finding out whether the very little added value
of the higher spatial resolution could eventually be attributed to the way how surface
CO2 fluxes are implemented in WRF-Chem. I cannot recommend the publication of
this paper unless the paper is improved in these aspects.*

**Response**: We thank the reviewers for the valuable comments. We have clarified the
implementation of CMS fluxes into WRF. We have used no interpolation nor projection
tools but instead directly assigned CMS low-resolution fluxes to the high-resolution WRF pixels within the grid box. Scaling factors were applied only to compensate for coastal areas and small changes in total fluxes due to the pixel attribution between WRF and CMS. These adjustments remain small compared to the actual fluxes and fully preserve the flux distribution. We have clarified these different points in the main text.

**Detailed comments**

*Throughout the paper the language of the paper is imprecise which makes major parts of the paper hard to read. Already the abstract is a good example:*

*What is meant by 'fluxes' in the first sentence? Fluxes from the surface? Fluxes from which sources?*

We have clarified that point. "CO2 surface fluxes"

*What is meant by 'transport'? Long range transport?*

We have clarified that point. "atmospheric transport simulations"

*What is 'our' North American domain?*

We have clarified that point.

*A few further examples (by far not complete) are:*

*Page 1, last line: 'do not agree well': With what?*

*Page 2, lines 21 and 23: The expression 'curtains' is somewhat odd.*

**Response:** We have clarified the text as suggested. Please find the revised version with colored modifications.

*Page 6, first line: What does 'dried of water vapor' mean here (odd wording anyway)?*

**Response**: The GEOS-Chem model had an incorrect representation of mole fractions which included, in part, water vapor in the total mass of air. We had to remove the contribution of water vapor to the total air mass before coupling to the WRF model. We have clarified in the text: "after correcting for the presence of water vapor to obtain dry air mole fractions" .

*The meteorological driver of GEOS-Chem is GEOS-5 while meteorological boundary conditions for WRF-Chem are from ERA interim. In how far can differences between these meteorological drivers contribute to the differences in upper air wind fields and CO2 concentrations from WRF-Chem and GEOS-Chem. This should be analyzed in more detail.*

**Response**: We agree with the reviewer that we have coupled the mole fractions from CMS and not the actual mass fluxes of CO2 molecules through the boundaries. Therefore, driver data will play a significant role between the two models at daily to weekly time scales. While we acknowledge the fact that wind fields will vary between the two models, this question would lead to separating the impact of the WRF model physics from the driver data. This comparison would be worthwhile but we do not expect significant variations at seasonal time scales. Both systems (GEOS and ERA-I) assimilate similar meteorological observations, hence represent the mean seasonal wind in similar ways. We have investigated the impact of driver data in regional simulations of CO2 (Diaz-Isaac et al., 2018) and found that daily variations from different driver data

can be large, but we do not expect major differences at the seasonal scale. We also note that GEOS-Chem has its own model schemes to utilize the wind from GEOS-5 to transport CO2, which differs from WRF (advection, diffusion, convection schemes). This additional layer will confuse the comparison whether differences come from driver data or from the model physics. We have added text in the manuscript. "The two different re-analysis driver data used here might also cause differences in simulated XCO2 and CO2 mole fractions. No reconciliation was performed because both models re-interpret driver data to a certain extent (through advection and diffusion schemes). However, comparison of ERA-Interim and GEOS-5 driver data would potentially bring additional information about transport differences."

*The authors describe in much detail how they achieve mass conservation when deriving CO2 surface fluxes for the WRF-Chem simulation from the fluxes applied to GEOS-Chem. Does this result in surface fluxes smoothed to the GEOS-Chem grid, which are used as input for WRF-Chem? What CO2 emission patterns (anthropogenic, biogenic) are still resolved in the WRF-Chem simulation? It is not clear whether the emissions for the WRF-Chem simulations are really better spatially resolved than for the GEOS-Chem simulations. Eventually show emission input for WRF-Chem and for GEOS-Chem in the supplementary material.*

**Response**: Mass-conservation for the fluxes is due to mis-alignment of coastal fluxes, near water bodies, and to small mismatches when assigning a small WRF pixel to the corresponding large CMS pixel. No smoothing nor any deformation was applied when re-gridding the surface fluxes. The scaling factor is computed over the entire domain and applied to the entire domain. The spatial distribution of the fluxes remains identical in WRF and CMS. And all flux components from the CMS model are used to have identical surface fluxes in both models. We have clarified these points in the text (Method section).

*Page 6, line 2: 'We do not scale the diurnal cycle overlay.'? How large is the difference between the diurnal cycles of the emissions applied to GEOS-Chem and WRF-Chem?*

**Response**: To clarify, the same diurnal cycle of the fluxes is applied to both models. Only the scaling factor (correction after re-gridding) of the fluxes is kept constant over a month, instead of being calculated for each hourly flux map. The result is minor (few tenths of a percent) compared to scaling for all the 3-hourly fluxes. We have clarified the text.

*Section 2.4.2: Why is the averaging performed over such a big area?*

**Response**: The sparsity of GOSAT soundings forces us to select a wide area around the TCCON site. Ideally, this box would be limited to a few degrees, or based on synoptic-scale pressure fields. To have a sufficient number of GOSAT soundings, we extended the box to 6x12 degree in size. Other methods have been developed (Guerlet et al., 2013) but rely on simulated CO2 fields which vary depending on the model used. We clarified in the text.

*Page 9, lines 3-6: These sentences are hard to understand. This should be explained in more detail. Please mention also the magnitude of the differences between the individual GOSAT XCO2 soundings which are located within a single GEOS-Chem cell?*

**Response**: We have clarified the text. The sampling density of GOSAT remains very low for any given CMS grid cell. The number of soundings in a typical CMS grid cell varies greatly with an annual estimate, between 1 to 300 soundings per grid cell. Over Summer, we applied a threshold of 10 soundings per grid cell, or one every three days.

[Figure]

In other terms, differences within a grid cell would quantify day-to-day variations but rarely overlaps in the same day.

*Page 12, line 14: Why is convective transport of CO2 not included? WRF-Chem can handle convective vertical transport of atmospheric trace compounds.*

**Response**: We have explored in great details the problem of convection parameterization in WRF-Chem. Currently, the WRF-Chem code includes an offline parameterization of convective tracer transport based on precipitation rates. As we have carefully investigated, we have found no direct coupling between mass fluxes within the convection scheme and the chemistry code. We are currently working on that issue and will publish results in the coming months. We agree with the reviewer's comment that we could have used the offline coupling (conv_tr). In this simulation, we have turned on the convection scheme but no direct coupling to the chemistry scheme was used. We hope to fix that problem in future simulations.

*The conclusions must be extended. Currently they contain mostly a description of data availability and a short summary*

**Response**: We have modified significantly the conclusions, focusing on a description of our results which includes the comparison of modeled mixing ratios to satellite, aircraft and tower observations.